# Supporting mental well-being of healthcare workers using a mobile app: A mixed-methods feasibility study

Mehmet Yildirim[1], Timothy Carter[1], Holly Blake[1,2]*

1 School of Health Sciences, University of Nottingham, Nottingham, United Kingdom, 2 NIHR Nottingham Biomedical Research Centre, Nottingham, United Kingdom

* holly.blake@nottingham.ac.uk

## Abstract

Poor mental well-being is common among healthcare workers, affecting individual health, patient safety, and organisational performance. Mobile app-based self-care interventions are promising due to their accessibility, low cost, and ease of use. This study aimed to assess the feasibility of a self-monitoring mobile app called MYARKEO, to improve mental well-being among healthcare workers and healthcare trainees in the United Kingdom (UK). The study evaluated recruitment and retention rates, variability of key outcomes to inform a future randomised controlled trial (RCT), intervention engagement, barriers and facilitators to engagement, and potential refinements to the mobile app. A mixed-method feasibility RCT was conducted with two groups: an intervention group using MYARKEO to monitor mental well-being over 6 weeks, and a non-intervention control group. Data were collected at baseline and post-intervention and included the Warwick-Edinburgh Mental Well-being Scale (WEMWBS), the Depression Anxiety and Stress Scale (DASS-21), and the mHealth App Usability Questionnaire (MAUQ). Qualitative data were collected through semi-structured interviews (n = 13) and analysed using thematic analysis. Forty-nine participants (32 workers, 17 trainees; aged 18–60+) were included in the trial, with a 20.5% dropout rate. Daily app usage averaged 64.5%. Participants frequently monitored mood, sleep, food, and exercise. Interviews identified themes of "Usefulness," "Enablers of engagement," "Barriers to engagement," and "Suggested intervention improvements." This study demonstrates the feasibility of using a mobile app to monitor and promote mental well-being among healthcare workers and trainees. While app engagement was promising, challenges were identified, highlighting the need for refinements to the app's content, interface, and design for future trials.

**Data availability statement:** Datafiles containing questionnaire and app usage data are available from The University of Nottingham Research Data Management Repository (http://doi.org/10.17639/nott.7594).

**Funding:** The Ministry of National Education of the Republic of Türkiye funded this study. Funded under Law No. 1416 by the Republic of Türkiye. The funders did not play any role in the study design, data collection and analysis, decision to publish, or preparation of the manuscript.

**Competing interests:** The authors have declared that no competing interests exist.

## Introduction

Health is a continuum concept, ranging from illness (being well means somebody is not sick) to wellness, meaning optimal health, balance, and stability [1]. Mental well-being, a critical component of this continuum, refers to achieving the best state of mental health, encompassing physical, social, and emotional capacities to cope with life's challenges [2]. However, the mental well-being of healthcare workers is a growing concern globally [3–6]. This population faces disproportionately higher risks of mental health problems compared to the general population [7,8], with mental ill-health accounting for around 27.7% of all sickness absences in this workforce [9].

The coronavirus (COVID-19) pandemic escalated this problem by increasing stressors for healthcare workers, leading to more mental health issues or exacerbating existing mental health concerns [10]. In the UK, for example, The impact of COVID-19 on nurses (ICON) study identified high rates of stress and anxiety in frontline healthcare workers during the pandemic [11]. There is an urgent need for accessible, scalable, and effective interventions to support the mental well-being of healthcare workers and healthcare trainees (the next generation of healthcare workers), who face unique stressors in their professional environments [12,13].

Mobile health (mHealth) applications (i.e., apps) have emerged as promising tools for addressing these challenges due to their accessibility, convenience, and ability to provide real-time feedback [14]. Designed for smartphones and other mobile devices, these apps offer users convenient access to features and resources that support self-care efforts, including symptom tracking, medication reminders, goal setting, mindfulness exercises, and educational materials [15]. The two leading categories of mHealth apps focus on wellness management and disease management, with additional functionalities such as self-diagnostic tools, medication reminders, and electronic patient portals [16]. Notably, apps targeting mental well-being often emphasise mindfulness activities, offering training and guided exercises to enhance users' mental health awareness and coping strategies [17]. Despite the increasing use of apps for well-being, the feasibility, usability and acceptability of such tools has not been investigated with healthcare workers or healthcare trainees [18]. While recent studies have highlighted the utility of mobile apps in tracking mental health metrics and delivering tailored interventions [19,20], to our knowledge, this is the first study focusing on comprehensive lifestyle tracking and its impact on the mental well-being of current or next generation healthcare workers.

In this study, we explored the feasibility and acceptability of self-monitoring mental well-being using a mobile app (MYARKEO) designed for the general population. The MYARKEO app, designed in English, enables users to track various aspects of their mental health (all questions and possible answers shown in S1 Table). By generating a daily score based on user-reported data, the app provides insights tailored to individual needs, fostering self-awareness and behavioural change. The app incorporates principles of self-determination theory (SDT) and Reinforcement Theory (RF). SDT posits that individuals are motivated to engage in behaviours that satisfy their needs for autonomy, competence, and relatedness [21]. The app supports autonomy

by allowing users to customise their tracking, competence by providing daily scores, and relatedness through potential discussion of their progress with friends or family members. The RF suggests that behaviour is shaped by reinforcement [22], and the app uses daily scores to positively reinforce healthy habits. These theories provide a framework for understanding how the app may promote sustained engagement and mental well-being improvement through healthy behaviour change.

The primary aim of this study was to assess the feasibility of a self-monitoring-based mobile app called MYARKEO, to improve mental well-being among healthcare workers and trainees in the UK. The secondary aim was to qualitatively explore the mobile app's acceptability and usability through users' daily experiences, identifying barriers and facilitators to engagement with the monitoring.

## Methods

### Design

This was a two-arm prospective feasibility randomised controlled trial (RCT) with a nested qualitative study. The RCT comprised an intervention group (mobile app access) and a non-intervention control group. Reporting was guided by the Consolidated Standards of Reporting Trials (CONSORT) [23] for pilot and feasibility studies (S2 Table), the Checklist for Reporting Qualitative Studies (COREQ-32) [24] (S3 Table), and mobile health (mHealth) evidence reporting and assessment (mERA) checklist [25] (S4 Table). The intervention is described using The Template for Intervention Description and Replication Checklist (TIDieR) (S5 Table). The study received ethical approval from the University of Nottingham Faculty of Medicine and Health Sciences Research Ethics Committee (Ref: FMHS 172-0221) on 11 March 2021. All participants provided written informed consent. The study protocol was presented at the University of Nottingham Research and Knowledge Exchange Festival in June 2022 and was registered on protocols.io on 17 December 2025 (dx.doi.org/10.17504/protocols.io.5jyl8xre9v2w/v1).

### Participants and setting

Healthcare workers or healthcare trainees, ≥18 years of age, with access and ability to use a mobile app, from any healthcare related occupational group, working in the UK, were eligible to participate. The UK has well-established digital infrastructure, including widespread 4G/5G coverage, high smartphone penetration, and reliable electricity [26]. Participants were excluded if they were away from work or studies due to mental ill-health.

### Recruitment

Participants were recruited between 1st April 2021 and 15th November 2021. Recruitment was conducted through social media posts and e-mails among professional networks, including Twitter (now 'X'), Facebook, two digital newsletters produced by the UK Foundation of Nursing Studies, and the UK Florence Nightingale Foundation. Convenience and snowball sampling was applied. For context, recruitment occurred during the COVID-19 pandemic, during a period of intense pressure on the healthcare workforce, globally. Interested participants were sent the study participant information sheet (PIS) and online informed consent was subsequently obtained.

### Randomisation

Randomisation to the groups was undertaken by a researcher who was independent of the study team, using a computer-generated random sequence. The randomisation sequence was generated using permuted blocks of variable size, randomly varying between four and six participants, to ensure balance between groups. Due to COVID-19 social distancing restrictions at the time of the study, the independent researcher informed MY about the allocation online as each participant enrolled in the study.

## Sample size

There is no agreed sample size for a feasibility study, although based on previous literature [27,28], we aimed to recruit at least 40 participants.

## Intervention arm

Those allocated to the intervention gained access to a commercially developed mobile app called MYARKEO (accessed on the Apple Store and Play Store), which primarily supported participants to engage in daily monitoring of mental well-being over six weeks. The goal of the intervention is to promote participants' mental well-being by stimulating their self-awareness and self-care in their lifestyle.

MYARKEO functions as a standalone self-tracking tool stimulating participants to keep a daily record of their health-related behaviour, symptoms and mood. Participants are asked to track a range of physical lifestyle factors and mental well-being items, including daily mood, stress levels, anxiety, work-related stress, low energy, general worry, negative thoughts, dietary choices, sleep patterns, exercise routines, caffeine and alcohol consumption, smoking habits, medication usage, menstrual cycles, and menopause symptoms. The participants are asked to only track the data they want to monitor based on their personal preferences. Based on their answers they receive a daily score to reflect the level of their mental well-being (score range: 0–100). The score does not represent a diagnosis of depression, anxiety, or stress; rather it represents a basic measure of well-being with a higher score representing increased engagement with behaviours consistent with improved well-being, improved mood and decreased symptomology of poor well-being (i.e., stress and anxiety). The participant is also given tailored feedback via the app in the form of a breakdown of what has changed from the previous day. All the questions related to mental well-being can be found in S1 Table. MYARKEO screenshots are available here [29].

## Control arm

Those allocated to the control arm were sent an e-mail signposting them to a UK National Health Service (NHS) website containing information about mental well-being improvement. This website contains generic information that is widely accessible to the public and provides steps around how to improve mental well-being with 'do' and 'do not' instructions [30]. No further intervention was offered to the control arm participants.

## Outcomes

Primary outcomes focused on the feasibility of a future RCT, and the usability of the intervention as follows: (1) recruitment rate, (2) adherence to the intervention and control groups, (3) data collection procedures, (4) attrition rates and underlying reasons, and (5) the usability of the mobile app. The usability of the mobile app was evaluated using the mHealth App Usability Questionnaire (MAUQ) [31], specifically designed for mobile health-related apps. Intervention adherence was measured by daily mobile app usage data.

Secondary outcomes included the following: mental well-being, measured using the Warwick-Edinburgh Mental Well-being Scale (WEMWBS) [32]; depression, anxiety, and stress, measured using the Depression, Anxiety, and Stress Scale (DASS-21) [33]; and app usability, assessed using the mHealth App Usability Questionnaire (MAUQ) [31]. The WEMWBS is valid, reliable, and acceptable in adult populations across Europe and in minority ethnic groups [34]. The DASS-21 demonstrates good psychometric properties, including high internal consistency [35], strong test-retest reliability [35], and good construct validity [36] through correlations with established measures of depression, anxiety, and stress. The MAUQ has three subscales, all with high internal consistency reliability [31].

## Interviews

All intervention participants were invited to undertake a qualitative semi-structured interview exploring users' interactions with the mobile app and evaluating its perceived usability. The interview topic guide can be found in S1 Text.

## Data analysis

**Quantitative data analysis.** The quantitative analysis was predominantly descriptive, with a primary focus on determining feasibility by estimating recruitment, attrition, and non-compliance rates. At the end of the six-week study period, raw usage data from the mobile app were descriptively analysed (though frequency counts) to explore the frequency with which items (symptoms, behaviours) were monitored. The daily engagement rate was calculated by dividing the total days participants actively used the app by the total possible usage days (19 participants × 42 days = 798).

Additionally, the means and standard deviations of mental health and well-being outcomes by the group at baseline and post-intervention were calculated to indicate the likelihood of change in outcome measures (to inform a future trial). We determined a 95% confidence interval (CI) for differences in means of outcomes between groups and assessment of change following the intervention at the end of the trial. Sample size calculations for the future RCT were estimated using the intervention effect size and variance estimates from the immediate post-intervention change data for the selected outcome measure.

Demographic and intervention data were reported as means and standard deviations (SDs) for continuous parametric data, medians and ranges for continuous non-parametric data, and frequencies and percentages for categorical data. IBM SPSS Statistics (Version 26) [37] was used for data analysis.

**Qualitative data analysis.** Braun and Clarke's thematic analysis approach [38] was applied to the interview data. The study researcher (MY), trained in qualitative research methods, conducted and transcribed all the interviews, which helped with familiarisation with the data. Subsequently, MY double-read each transcription for further familiarisation and development of initial impressions and identification of recurring patterns [38]. The transcriptions were then uploaded to NVivo v12 [39] for the initial coding and subsequent inductive grouping into potential themes [38]. This involved identifying patterns and relationships between codes to form overarching themes that capture the essence of the data [40]. To ensure robust analysis, the initial codes and themes were revisited and shared with the research team for their opinions and recommendations to ensure that the themes were coherent and well-supported by the data [41]. After a series of discussions, the main and sub-themes were named and defined.

Finally, the findings were triangulated and reported in conjunction with the quantitative results. Triangulation was done by integrating qualitative data from semi-structured user interviews with quantitative data collected from the mobile app use findings and survey responses. Data about reasons for dropouts, consistency of the use, acceptability and usability of the app were triangulated.

## Results

### Recruitment

Forty-nine healthcare workers, and healthcare trainees were recruited, three did not participate. Participants (n=46) included 17 nurses, 17 healthcare trainees (of which nine were registered on courses in nursing, four psychology, two physiotherapy, and two medicine), three physiotherapists, two support workers, two primary care counsellors, and one midwife, one physician, one psychotherapist, one psychologist, and one dentist. Recruitment rate was approximately eight participants per month. Details of participant flow are presented in Fig 1. Recruitment was stopped after reaching the target sample size.

### Demographics

Demographic data included 46 participants as three dropped out of the study after signing the consent form so were not randomised. Most participants were woman (84.8%, n = 39), aged 21–40 years (78.2%, n = 36), and two-thirds were White (67.3%, n = 31). Participants came from diverse occupational backgrounds although the majority were either nurses (36.9%, n = 17) or healthcare trainees (36.9%, n = 17) (See Table 1).

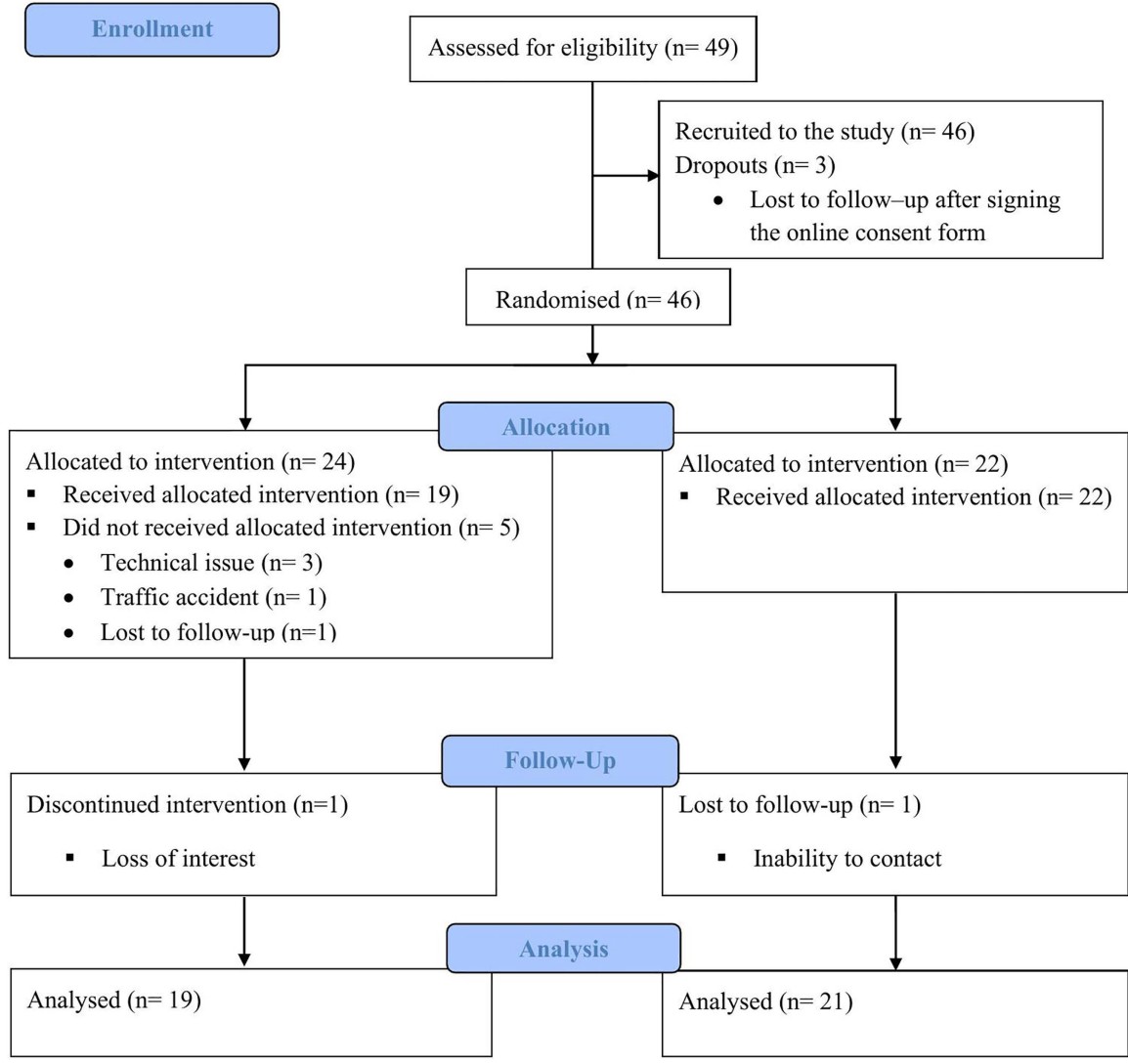

**Fig 1. CONSORT 2010 flow diagram.**

## Feasibility outcomes

All four feasibility criteria are summarised in Table 2.

Ten of 49 participants dropped out of the study for several reasons, including a technical problem, inability to contact, losing interest, and a traffic accident. The dropout rate was 20.5%, and the retention rate was 79.5%. The dropout rate was higher in the intervention group (25%, n = 6) than the control group (4%, n = 1). In the intervention group, reasons for dropout included technical issues (n = 3), a traffic accident (n = 1), inability to contact (n = 1), and loss of interest (n = 1). The dropout in the control group was due to an inability to establish contact.

The primary technical difficulty reported by Android users was an error message stating 'invalid email or password,' which could not be resolved. As a result, three participants were unable to access the mobile app after providing consent and completing the baseline survey.

**Table 1. Sample characteristics.**

| Characteristics (n = 46) | Total (n, %) | Intervention (n, % within group) | Control (n, % within group) |
|---|---|---|---|
| Age | | | |
| 18-20 | 1 (2.2%) | 0 | 1 (4.5%) |
| 21-30 | 22 (47.8%) | 9 (37.5%) | 13 (59.1%) |
| 31-40 | 14 (30.4%) | 7 (29.2%) | 7 (31.8%) |
| 41-50 | 5 (10.9%) | 4 (16.7%) | 1 (4.5%) |
| 51-60 | 1 (2.2%) | 1 (4.2%) | 0 |
| 60+ | 3 (6.5%) | 3 (12.5%) | 0 |
| Gender | | | |
| Man | 5 (10.9%) | 1 (4.2%) | 4 (18.2%) |
| Woman | 39 (84.8%) | 21 (87.5%) | 18 (81.8%) |
| Non-Binary | 2 (4.3%) | 2 (8.3%) | 0 |
| Ethnicity | | | |
| White | 31 (67.4%) | 16 (66.7%) | 15 (68.2%) |
| Mixed | 2 (4.3%) | 1 (4.2%) | 1 (4.5%) |
| Black/Black British | 1 (2.2%) | 1 (4.2%) | 0 |
| Asian/Asian British | 10 (21.7%) | 4 (16.7%) | 6 (27.3%) |
| Chinese and others | 2 (4.3%) | 2 (8.3%) | 0 |
| Occupational Group | | | |
| Nurse | 17 (37.0%) | 9 (37.5%) | 8 (36.4%) |
| Healthcare trainee | 17 (37.0%) | 6 (25.0%) | 11 (50.0%) |
| Physiotherapist | 3 (6.5%) | 3 (12.5%) | 0 |
| Support worker | 2 (4.3%) | 2 (8.3%) | 0 |
| Primary care counsellor | 2 (4.3%) | 2 (8.3%) | 0 |
| Midwife | 1 (2.2%) | 0 | 1 (4.5%) |
| Medicine | 1 (2.2%) | 1 (4.2%) | 0 |
| Psychotherapist | 1 (2.2%) | 0 | 1 (4.5%) |
| Psychologist | 1 (2.2%) | 1 (2.2%) | 0 |
| Dentist | 1 (2.2%) | 0 | 1 (4.5%) |

*Trainees include nursing (n = 9), psychology (n = 4), physiotherapy (n = 2), and medicine (n = 2).

**Table 2. Feasibility criteria.**

| Feasibility criteria | Study figures |
|---|---|
| Number of participants | 49 |
| Dropout rate | 20.5% |
| App usage | 64.5% |
| Retention rate | 79.5% |

## Intervention engagement

Participants were asked to use the mobile app for 42 days (six weeks). Daily mobile app usage declined over time as the days progressed. Nineteen of 24 (79.1%) participants started to use the app. Fig 2 presents how frequently 19 participants used the mobile app on a weekly basis. Across all users, the overall daily usage rate was 64.41% (514/798 days), and the total missing rate was 35.59% (284/798 days). Daily usage ranged between four and 42 days, (mean = 27.05). The items that were monitored daily are detailed in Supplementary files (S1 Fig).

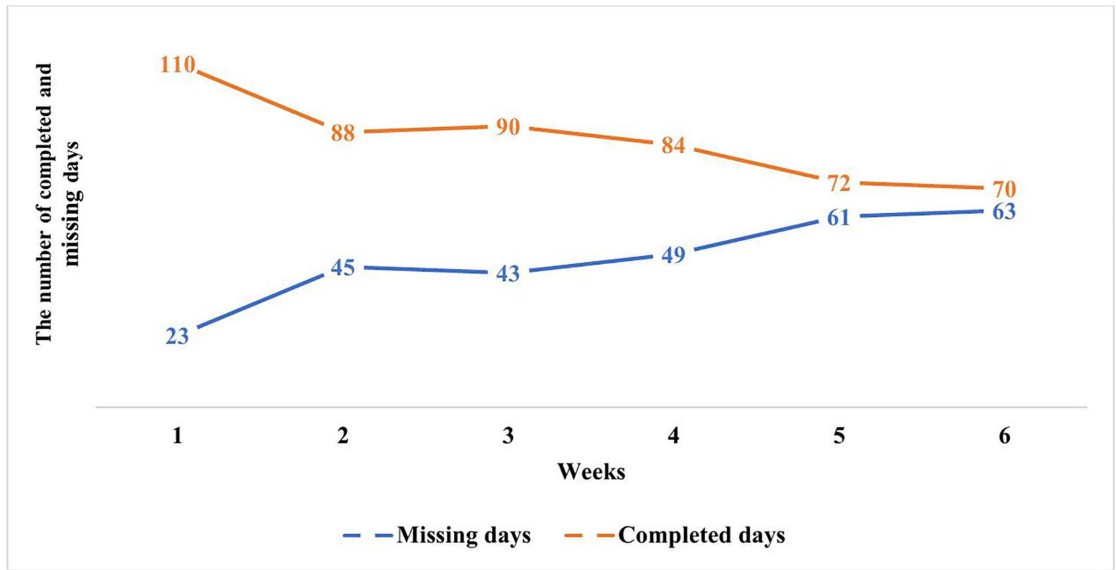

**Fig 2. The pattern of completed and missing days on a weekly basis (n = 19).**

## Usability of the mobile app

Eighteen participants completed the MAUQ questionnaire, with 17 included in the analysis as one was excluded for answering fewer than 15 questions [42]. Table 3 summarises the mean scores, standard deviations, and ranges for total and sub-scores. The total mean score of 5.2 on the Likert scale indicated "Good" overall usability and acceptability. Participants rated ease of use as "Excellent" (6.2), but satisfaction with the interface ("Poor", 4.8) and the app's usefulness for mental well-being as ("Poor", 4.1) were lower.

## Assessment of mental well-being, depression, stress, and anxiety

Table 4 compares mean scores for well-being, depression, anxiety, and stress between the control and intervention groups. Well-being levels remained moderate before and after the intervention in both groups. The control group showed a slight increase in well-being scores (from 47.64 [SD = 7.4] to 50.05 [SD = 9.1]), while the intervention group showed a slight decrease (from 49.42 [SD = 7.7] to 48.83 [SD = 10.8]). However, neither change was statistically significant (control: p = .22, intervention: p = .97). Notably, in the intervention group, participants who dropped out had a higher mean baseline well-being score (51) than the overall baseline (49.42) and post-intervention mean scores (48.83). Given the DASS-21 scale, scores for depression, anxiety, and stress in both groups remain in the mild-to-moderate range, with no significant reductions post-intervention.

**Table 3. The mobile app usability findings.**

| Items | Likert-scale means | Mean scores (SD) | Range |
|---|---|---|---|
| Total score | 5.2 – Good | 84.17 (25.5) | 35-123 |
| Ease of use | 6.2 – Excellent | 28 (7.7) | 9-35 |
| Interface and satisfaction | 4.8 – Poor | 33.83 (11.6) | 15-49 |
| Usefulness | 4.1 – Poor | 22.33 (9.5) | 9-41 |

**Table 4. The mean scores of well-being, depression, anxiety, and stress within the groups.**

|  | Pre-intervention | Post-intervention | p-value* |
|---|---|---|---|
| **Control group** |  |  |  |
| WEMWBS | 47.64 (7.4) | 50.05 (9.1) | .22 |
| Depression | 9.27 (6.3) | 9.24 (5.2) | .94 |
| Anxiety | 8.18 (6.8) | 8.67 (6.9) | .77 |
| Stress | 14 (8.3) | 13.81 (6.8) | .96 |
| **Intervention group** |  |  |  |
| WEMWBS | 49.42 (7.7) | 48.83 (10.8) | .97 |
| Depression | 7.58 (6.6) | 8.78 (8.8) | .61 |
| Anxiety | 6.33 (6.4) | 6.11 (7.7) | .54 |
| Stress | 13.92 (7.2) | 13.67 (8.7) | .64 |

*Significant at $p < .05$.

### Help-seeking behaviour

Participants were asked if they sought mental health support during the intervention to assess whether the mobile app influenced help-seeking behaviours. In the intervention group, 22.2% reported seeking mental health-related help (e.g., from a general practitioner, coach, or friends), broadly comparable to 23.8% in the control group (e.g., from friends, supervisors, or colleagues) (Table 5).

### Contamination

To evaluate potential contamination, participants were asked about their use of other mental health-related mobile apps during the study. Participants were not restricted from using such apps. Most participants did not use other apps, with 94.4% in the intervention group and 90.5% in the control group reporting no additional mental health related app usage. Only one participant (5.6%) in the intervention group and two participants (9.5%) in the control group reported using another app (Table 5).

### Qualitative findings

Interviews were conducted between 24th June 2021 and 22nd February 2022, lasting between 11 and 37 minutes (mean: 19 minutes). Thirteen participants were interviewed, aged 21–60+ including ten women, two non-binary individuals, and one man. Interviewees were from a range of occupations, including nurses (n = 5), healthcare trainees (n = 5), a physiotherapist (n = 1), a primary care counsellor (n = 1), and a psychologist (n = 1).

Four main themes were generated: "Usefulness," "Enablers of engagement," "Barriers to engagement," and "Suggested intervention improvements" along with 11 sub-themes. Detailed themes, sub-themes, and representative quotes are presented in S6 Table.

**Table 5. Help-seeking behaviour and contamination.**

| Group | Behaviour | Yes n (%) | No n (%) | Total n (%) |
|---|---|---|---|---|
| Intervention | Help-seeking behaviour | 4 (22.2) | 14 (77.8) | 18 (100) |
|  | Other app use | 1 (5.6) | 17 (94.4) | 18 (100) |
| Control | Help-seeking behaviour | 5 (23.8) | 16 (76.2) | 21 (100) |
|  | Other app use | 2 (9.5) | 19 (90.5) | 21 (100) |

## Theme 1: Usefulness

**Impacts on behaviours.**  The app facilitated awareness and positive changes in habits like sleep, exercise, and smoking. Participants highlighted its user-friendly design, ease of use, and ability to track trends over time. For instance, some improved fitness routines or reduced smoking by monitoring patterns. Less common aspects tracked included caffeine, alcohol, nicotine use, menopause symptoms, and medication, with participants valuing reminders for medication and attention to menopause symptom tracking. Despite these benefits, a few participants reported no behavioural changes, citing reasons such as the app's lack of advice or their personal circumstances.

*"I thought it was really easy to use, definitely. It doesn't take long, so it's not like a burden on your day."* Participant 5, Healthcare trainee, Woman.

*"I've been thinking about giving up smoking. I noticed I pretty much smoked than my usual amount… Then, I have stopped smoking as much as I can."* Participant 3, Primary care counsellor, Non-binary.

**Impacts on mental well-being.**  While some participants found the app beneficial for self-reflection and understanding emotional triggers, many reported no perceived direct improvement on their mental well-being. Insights gained included better understanding the connections between caffeine intake and anxiety or the effects of long work shifts. However, participants emphasised that the app lacked actionable advice, and improvements in mental well-being often required more time or external support. Those who already felt mentally well saw limited impact from the app on their own well-being per se but appeared to gain insight from the app as a tracking tool, nevertheless.

*"I think not. Probably not a direct impact as such, but it's made me realise different things that may affect your mental health."* Participant 6, Nurse, Woman.

## Theme 2: Enablers of engagement

**Increased awareness of mental well-being.**  Participants reported that the app helped them gain insights into their thoughts, emotions, and behaviours. By tracking and visualising data, they identified patterns and areas for improvement, which motivated consistent use. Features like regular tracking, visual representations, and self-reflection improved a sense of control and responsibility over their mental well-being.

*"It helped me to be more mindful of my mental well-being."* Participant 11, Healthcare Trainee, Woman.

**Practical strategies.**  Participants developed strategies to support consistent app use. One of the strategies involved placing the app icon prominently on the home screen of their mobile phones. This simple adjustment significantly increased its visibility and provided users with easy access whenever they needed to engage with the app, minimising the likelihood of forgetting to use it.

*"It was on my home page so I tended to see it there, but I think the future I would set, and I would get the notifications to remind me more."* Participant 9, Nurse, Woman.

Another important strategy was the setting of alarms as reminders. This approach served as a compensatory measure for the absence of in-app notifications, which can sometimes be overlooked or ignored. These alarms helped them establish a routine for interacting with the mobile app.

*"I tried to set the same time every day to complete the app."* Participant 11, Healthcare Trainee, Woman.

**External motivators.** There was a general sense that external prompts and factors that stimulated use the mobile app were important. The most frequently reported external motivator was the weekly email reminder sent by the study researcher. These emails served as a gentle nudge, reminding participants of their involvement in the study and the potential benefits of using the app. Other external motivators included the quick and easy input process for answering daily questions, the personalised flexibility to complete these questions at any time of day, and the option to enter data from the previous day, if they had not logged it. Furthermore, knowing they were part of a study, feeling observed, and believing in the app's benefits, motivated participants to keep using it.

*"The emails helped me to remember to use the app."* Participant 8, Healthcare Trainee, Woman.

*"I think it's where I was conscious that it was just a six-week trial. You know, to make sure that was done for six weeks. I think that probably made me focus on it more."* Participant 1, Nurse, Woman.

### Theme 3: Barriers to engagement

**Personal challenges.** The most common reported challenges were forgetfulness and being busy during the working day. These two challenges were interrelated, as participants expressed that being busy often led to them forgetting to use the app. Many participants did not prioritise using the app, especially when engaging in work-related tasks. Night shifts also posed a barrier, as participants who worked during the night were often too tired to log their data.

*"I used it when I remembered to use it if I'm honest. And I think I'd probably use it on days when I was doing really well. So, days when I noticed I was really busy and not feeling great. And then I did forget to use it. Actually, because it just went out of my mind."* Participant 13, Nurse, Woman.

Other personal difficulties included a preference for more detailed information tracking, and/or dissatisfaction with the app's questions or scoring system. Some participants wanted to input precise data, such as their exact sleep hours, but found the available options insufficient. For example, the ranges for the sleep question within the app (0–3, 4–6, etc.) confused participants who slept between 3–4 hours. Mood-related questions also caused dissatisfaction, as participants felt their mood fluctuated throughout the day and the app's questions were too general. Participants suggested rephrasing questions to reflect specific times of day or provide more precise prompts.

*"There's a few bits that I found a bit confusing, like the sleep bit—it was not three to four; it was four to six. I had a problem if I slept between three and four hours. Do I put it in the 0–3 range or the 4–6?"* Participant 18, Nurse, Woman.

**Motivational challenges.** Initial motivation was high, as participants believed the app would help to improve their well-being. However, motivation decreased over time when immediate results were not seen, causing some to lose interest and stop using the app. Other demotivating factors included lack of perceived improvement, and lack of advice on addressing problems. Participants noted that while the app helped them to self-monitor their behaviours, it did not provide actionable solutions for specific issues like stress or anxiety.

*"I think feeling down helped me (to use the app) because I was looking for something to keep myself up. But afterwards I had tough days. And the app wasn't enough to keep my motivation up. That's why I probably stopped it slowly. I ended up not using afterwards."* Participant 2, Physiotherapist, Woman.

 

**Technical challenges.** Technical challenges also emerged as a barrier to consistent app use. Participants encountered some minor issues such as software bugs, which included unresponsive questions, delayed saving of data, and daily scores not loading. Those bugs were resolved by restarting the app, but these interruptions still affected the user experience.

### Theme 4: Suggested intervention improvements

**Work-oriented improvements.** Participants highlighted the need for the app to better address the unique challenges of healthcare workers, such as stressful environments, long shifts, and neglect of self-care. They suggested adding personalised tools to monitor work-related aspects like communication with colleagues, shift patterns, hydration, and break usage. Break reminders and tracking were also recommended to promote recovery during work hours. Participants also wanted more options for absence tracking, including annual leave, to monitor mental well-being during breaks from work.

**Content improvements.** Participants suggested incorporating actionable advice and educational content to help users address identified well-being challenges. Specifically, they requested links to articles, videos, and brief tips, as well as explanations clarifying the relationship between behaviours and mental health. Push notifications were frequently recommended to mitigate forgetfulness and promote consistent app use:

*"I forgot. It wasn't a reminder on there. Maybe an option for a bit of a reminder."* (Participant 9, Nurse, Woman)

Additional proposals included the integration of peer support features to enhance motivation through shared experiences, and goal-tracking functions to enable visualisation of progress over time.

**Design improvements.** Participants expressed a desire for a more user-friendly, customisable interface. The current green-and-black colour scheme was perceived as dark and even 'masculine' by some. Brighter colours and personalised themes were suggested to make the app more inclusive. Adding interactive elements like emoji, animations, motivational quotes, and gamification features was proposed as a mechanism for enhancing engagement and enjoyment.

*"For me, making it seems like fun and bit of play... I would do more."* Participant 18, Nurse, Woman.

## Discussion

To our knowledge, this is the first study to evaluate the feasibility of a mobile app enabling comprehensive lifestyle tracking (e.g., mood, sleep, diet, exercise) for mental well-being among healthcare workers and healthcare trainees. While prior research has examined mental health apps targeting single domains (e.g., mindfulness [17] or symptom tracking [21]), none have combined multidimensional self-monitoring with real-time feedback for this high-risk population. Beyond assessing feasibility, this study also explores the app's usability, acceptability, and user experiences – critical factors for optimising engagement and implementation in healthcare settings.

Overall, our findings indicate that a comprehensive lifestyle-tracking app is feasible for use in this population, with encouraging engagement (64.5%) and retention (79.5%) rates. Nonetheless, technical difficulties, declining usage over time, and only moderate perceived usefulness highlight the need for refinement of app content and design prior to large-scale implementation.

Recruitment of 49 participants within six months met recommended sample size ranges [27,28]. The overall dropout rate (20.5%) was within acceptable ranges [43] and lower than rates reported in other digital intervention studies [44,45]. However, the intervention group had a higher dropout rate (25%) compared to the control group (4%), primarily due to a major technical issue affecting Android users (of which there were more in the intervention group). Resolving such technical issues could significantly reduce dropouts, as highlighted in similar studies [46,47].

The app achieved a 64.4% daily usage rate (mean = 27.05 active days), demonstrating stronger adherence than both: (1) comparable mHealth interventions for healthcare populations [48,49] and (2) the 6.38–23.07 mean active days reported in a meta-analysis of 92 mental health app RCTs [50]. This higher usage rate is likely due to the app's brief daily assessments and visual feedback, which align with Reinforcement Theory [22]. However, a declining usage pattern emerged over time, attributed to a perceived lack of benefits, technical issues, and healthcare workers' busy work schedules. Interviews revealed that reminders, such as notifications, could improve adherence, and this has been found elsewhere [51,52].

Although a feasibility study does not aim to examine effectiveness of the intervention, pre- and post-intervention data provide descriptive insights into potential changes in measures to inform the design of a future trial. At baseline, participants in both intervention and control groups reported moderate well-being levels (WEMWBS), consistent with the general UK population's well-being scores [53,54]. Although a feasibility study does not focus on determining effectiveness, substantial changes in well-being were not observed post-intervention, although this is potentially due to the small sample size [55] and a "ceiling effect," where moderate baseline scores may have limited any detectable improvements [56].

Interestingly, the control group showed a greater increase in well-being scores than the intervention group. This may be partly explained by attrition bias [57] in the intervention group. Dropouts in that group had higher baseline well-being scores than both completers and the post-intervention average, potentially skewing results. As noted earlier, attrition in the intervention group was linked to app usage on Android phones with login issues. Additionally, there were no significant differences in DASS-21 scores for depression, anxiety, or stress between groups or pre- and post-intervention.

The mHealth App Usability Questionnaire (MAUQ) was used to evaluate the mobile app, showing high overall usability and an excellent "ease of use" score. However, lower scores for "interface and satisfaction" and "usefulness" were noted. Interview insights highlighted reasons for these lower scores, including the absence of personalised advice, helpful links, reminders, and notifications, reducing engagement and perceived usefulness. A masculine interface design was also cited as a potential barrier. Literature supports that user engagement is boosted by feedback, notifications, and diverse, user-friendly design [58,59]. Without these features, participants reported limited motivation and perceived benefits, reducing the app's long-term effectiveness [60]. However, it should be noted that this study used an early version of the MYARKEO app, and a later version with updated content and interface was released during the intervention period. For consistency of experience among our participants, we continued with the same version for the study duration, noting that some of the reported issues may already have been resolved in the updated version.

The interviews explored participants' experiences with the mobile app for monitoring mental well-being. Participants appreciated the apps' ease of use, portability, and real-time data entry, which supported consistent use. They reported increased autonomy and self-control, aligning with Self-Determination Theory (SDT), which highlights autonomy, competence, and relatedness as key to intrinsic motivation [21]. The app was perceived as more beneficial for supporting habit formation (e.g., diet, exercise, sleep) than improving mental health-related symptoms. Behavioural changes are often more tangible, while mental well-being improvements involve complex, gradual changes in attitudes and emotions, making them harder to measure [61]. The app achieved a 64.4% engagement rate, which is higher than that observed in similar studies [48,49]. This was likely due to several perceived positive factors relating primarily to the app increasing participants' self-awareness and it being perceived to be motivational through the daily scoring system. The app's features enabled participants to identify and monitor areas for improvement, reinforcing healthy behaviours through positive feedback, as suggested by Reinforcement Theory [22]. Participants also employed individual strategies (e.g., placing the app icon prominently, setting alarms) to enhance engagement. The app's design, enabling quick data input and self-reflection through visualised progress further supported sustained use.

## Study strengths

This mixed-methods feasibility study is the first to evaluate a mobile app-based intervention for promoting mental well-being among healthcare workers and trainees. The app adopted a holistic approach by monitoring both physical and

mental health indicators. The qualitative component provided novel insights into participants' experiences and identified factors that contributed to engagement, including the app's scoring system, brief daily usage, and flexible options for tracking well-being. Engagement rates exceeded those typically reported in comparable studies, suggesting that these design features may have enhanced participation.

Participants were uniquely asked about their use of other well-being apps to account for potential confounding factors, establishing a precedent for greater transparency in future research. In addition, the study adhered to updated guidelines for reporting WEMWBS outcomes and incorporated novel descriptive metrics, strengthening methodological rigour.

### Study limitations

The study was conducted during the COVID-19 pandemic and adapted to social distancing requirements through online interactions and social media recruitment [62]. While effective, this approach may have reduced sample representativeness by excluding individuals not active on social media and skewing participation toward younger adults. Although sample demographics broadly aligned with NHS workforce patterns – particularly the higher proportion of female employees – other genders and older age groups may have been under-represented. Recruitment metrics could not be fully calculated due to the open accessibility of social media posts, limiting assessment of platform-specific efficacy. Reliance on digital recruitment alone may therefore constrain participant diversity, underscoring the need for traditional recruitment methods to improve generalisability in future research.

Although not directly observed, snowball sampling may have introduced bias through potential contamination between control and intervention groups, as participants could have shared experiences. Another limitation was the inability to compare findings with other studies using the MAUQ questionnaire, as this remains a relatively new instrument for assessing mobile app usability. Design constraints may also have reduced user engagement. Specifically, the absence of reminders and push notifications (due to financial limitations) likely limited sustained daily use. Weekly email reminders were implemented as a substitute; however, these may be less effective than interactive approaches such as in-app notifications.

### Conclusion

This study is the first to evaluate the feasibility and daily usage patterns of a mobile app designed to promote mental well-being among healthcare workers and trainees. Findings indicate that such an app is feasible and acceptable, with positive outcomes across key aspects of randomised controlled trial methodology, including recruitment, randomisation, intervention delivery, questionnaire administration, and participant retention. Results from the MAUQ, qualitative interviews, and user engagement data suggest that participants view these interventions favourably and that mobile apps can support the development of positive daily habits, potentially contributing to long-term improvements in mental well-being.

Nevertheless, the effectiveness of these interventions must be confirmed in large-scale randomised trials. To enhance engagement, design improvements are needed, such as push notifications, goal-setting features, and tailored content – including work-specific tools like hydration reminders and shift monitoring.

Given the rising prevalence of mental ill-health among healthcare professionals worldwide, this intervention aligns with national and international efforts to strengthen workforce well-being, with potential implications for care quality and patient safety. Digital solutions offer accessibility, cost-effectiveness, and practicality, making mobile apps a promising approach for mental well-being promotion. Because mobile phones are ubiquitous, these findings are relevant across diverse healthcare settings and countries.

### Supporting information

**S1 Table. All mental well-being related questions and possible answers in the mobile app.**
(DOCX)

**S2 Table. CONSORT 2010 checklist of information to include when reporting a pilot or feasibility trial.**
(DOCX)

**S3 Table. COREQ-32 checklist.**
(DOCX)

**S4 Table. Mobile health (mHealth) evidence reporting and assessment (mERA) checklist.**
(DOCX)

**S5 Table. The Template for Intervention Description and Replication Checklist (TIDieR).**
(DOCX)

**S6 Table. Themes and sample quotes.**
(DOCX)

**S1 Text. Interview topic guide.**
(DOCX)

**S1 Fig. Daily mobile app usage with missing days.**
(TIF)

## Acknowledgments

The authors thank Laura Bedford for contribution to discussions around the intervention in the early stages of the study, and Caner Varol for generating the randomisation sequence and informing the study researcher about group allocation. We thank Jana Dowling for developing the mobile app and providing permission for its use in this study. The mobile app (MYARKEO) adhered to the General Data Protection Regulation (GDPR). For the purposes of data protection legislation, MYARKEO is the data controller of user's personal data. MYARKEO is registered with the Information Commissioners Office in the UK with reference number A8485743. The participants accessed MYARKEO through the Apple Store or Play Store. The app developer had no role in study design, data collection and analysis, decision to publish, or preparation of the manuscript.

## Author contributions

**Conceptualization:** Mehmet Yildirim, Timothy Carter, Holly Blake.

**Data curation:** Mehmet Yildirim, Timothy Carter, Holly Blake.

**Formal analysis:** Mehmet Yildirim, Timothy Carter, Holly Blake.

**Funding acquisition:** Mehmet Yildirim.

**Investigation:** Mehmet Yildirim, Timothy Carter, Holly Blake.

**Methodology:** Mehmet Yildirim, Timothy Carter, Holly Blake.

**Project administration:** Mehmet Yildirim.

**Supervision:** Timothy Carter, Holly Blake.

**Writing – original draft:** Mehmet Yildirim, Timothy Carter, Holly Blake.

**Writing – review & editing:** Mehmet Yildirim, Timothy Carter, Holly Blake.

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
