## [Decision Letter · Decision Letter 0]

27 Jul 2025

Dear Dr. Blake,

Thank you for submitting your manuscript to PLOS ONE. After careful consideration, we feel that it has merit but does not fully meet PLOS ONE’s publication criteria as it currently stands. Therefore, we invite you to submit a revised version of the manuscript that addresses the points raised during the review process.

We look forward to receiving your revised manuscript.

Kind regards,

Najmul Hasan, PhD

Academic Editor

PLOS ONE

Journal Requirements:

3 .If the reviewer comments include a recommendation to cite specific previously published works, please review and evaluate these publications to determine whether they are relevant and should be cited. There is no requirement to cite these works unless the editor has indicated otherwise.

Reviewers' comments:

Reviewer's Responses to Questions

**Comments to the Author**

1. Is the manuscript technically sound, and do the data support the conclusions?

Reviewer #1: Partly

Reviewer #2: Partly

2. Has the statistical analysis been performed appropriately and rigorously?

Reviewer #1: Yes

Reviewer #2: Yes

3. Have the authors made all data underlying the findings in their manuscript fully available?

Reviewer #1: Yes

Reviewer #2: Yes

4. Is the manuscript presented in an intelligible fashion and written in standard English?

Reviewer #1: No

Reviewer #2: Yes

Reviewer #1: This article aims to assess the feasibility of a self-monitoring mobile app called MYARKEO, to improve mental well-being among healthcare workers and healthcare trainees in the United Kingdom (UK) .

First, in the introduction section, authors mentioned "Despite the increasing use of apps for wellbeing, the feasibility, usability and acceptability of such tools has not been investigated with healthcare workers or healthcare trainees. ” This section should be rewritten because the author(s) only need to clearly point out the theoretical basis and the reason of using these theories in this part.

Second, the literature review section is deficient in several key areas. It fails to sufficiently address recent scholarly works pertinent to the topic in the following respects:

1. “Chatbot as an emergency exist: Mediated empathy for resilience via human-AI interaction during the COVID-19 pandemic”

2. “Disentangling composite influences of social connectivity and system interactivity on continuance intention in mobile short video applications: The pivotal moderation of user-perceived benefits”

3. “Determining influence of service quality on user identification, belongingness, and satisfaction on mobile social media: Insight from emotional attachment perspective”

4. “How multidimensional benefits determine cumulative satisfaction and eWOM engagement on mobile social media: Reconciling motivation and expectation disconfirmation perspectives”

5.“Struggling or Shifting? Deciphering potential influences of cyberbullying perpetration and communication overload on mobile app switching intention through social cognitive approach”

6. “Determining the influence of depressive mood and self-disclosure on problematic mobile app use and declined educational attainment: Insight from stressor-strain-outcome perspective”

7�“Deciphering dynamic effects of mobile app addiction, privacy concern and cognitive overload on subjective well-being and academic expectancy: The pivotal function of perceived technostress”

Third, more details about the method are needed to add in this current article. Language needs polishing; many sentences are not very smooth

Fourth, according to the results, some findings were proposed. Combined with related work, the authors should accentuate their contributions in the conclusion section of this paper.

Finally, while the paper is well-structured, there are still some issues with the clarity of expression and language, undermining its readability. Thus, the authors should ensure that a revised version of the manuscript is thoroughly proofread.

Reviewer #2: Thank you for the opportunity to review your manuscript on MYARKEO. The study explores a timely topic, and the intent behind the intervention is promising. However, the manuscript would benefit from the following improvements:

1. Methodological Considerations

Technical Issue Specificity: In Theme 3: Barriers to engagement – Technical Challenges, the mention of “software bugs” is too general. Since the app is central to the study, it would be helpful to describe the types of bugs encountered and their impact. Login issues on Android and the lack of a password reset feature appear to have contributed to attrition. Given that the app was updated during the study, please clarify whether these problems were resolved and why participants were not encouraged to update, as updates typically address such usability concerns.

Engagement Calculation: The reported 64.41% engagement rate (514/789 days) lacks detail. Please clarify how this was calculated over the 6-week period, including definitions of engagement and the denominator used.

Tracking Frequency: Please specify how often participants used the app each day. This data is often available from app analytics. Encouraging more frequent use may have yielded richer insights, as prior research (e.g., Myin-Germeys et al., 2009) shows the importance of capturing intra-day fluctuations in mental health using Ecological Momentary Assessment (EMA).

2. Inconsistencies

Usability Ratings: The usability section rates ease of use as “excellent,” yet interface satisfaction and usefulness are rated “poor.” This contradiction should be clarified.

Participant Feedback vs. Quantitative Ratings: In Theme 1, participants are quoted as praising the app’s design and ease of use, yet this conflicts with the MAUQ ratings. Clarifying this discrepancy would improve interpretive consistency.

Dropout and Well-being Scores: The "assessment of mental well-being, depression, stress and anxiety" section mentions that participants who dropped out had higher baseline well-being scores than those who completed the study. This pattern appears to undermine the conclusion that the app supports positive behavior change or improved well-being, and should be discussed.

3. Reporting Clarity

Repetition: The phrase “expectation of immediate benefits” is repeated in Theme 3 – Motivational Challenges and could be consolidated.

Comparative Claims: The claim that engagement was higher than in similar studies needs supporting references to validate the comparison.

The study explores an important area of interest. Addressing the issues above will improve the manuscript’s clarity, methodological transparency, and alignment between data and conclusions.

**Do you want your identity to be public for this peer review?** For information about this choice, including consent withdrawal, please see our Privacy Policy

Reviewer #1: No

Reviewer #2: **Yes:**  Aishwarya Murali

---

## [Author Response · Author response to Decision Letter 1]

21 Aug 2025

Reviewer 1

This article aims to assess the feasibility of a self-monitoring mobile app called MYARKEO, to improve mental well-being among healthcare workers and healthcare trainees in the United Kingdom (UK).

First, in the introduction section, authors mentioned "Despite the increasing use of apps for wellbeing, the feasibility, usability and acceptability of such tools has not been investigated with healthcare workers or healthcare trainees.” This section should be rewritten because the author(s) only need to clearly point out the theoretical basis and the reason of using these theories in this part.

Response: Many thanks for this comment. We have checked the introduction and revised the flow of the paragraphs.

We have added a section into the end of paragraph 3 (line 61) in which we mention mobile app-based mental wellbeing interventions. This improved the flow.

Second, the literature review section is deficient in several key areas. It fails to sufficiently address recent scholarly works pertinent to the topic in the following respects:

1. “Chatbot as an emergency exist: Mediated empathy for resilience via human-AI interaction during the COVID-19 pandemic”

2. “Disentangling composite influences of social connectivity and system interactivity on continuance intention in mobile short video applications: The pivotal moderation of user-perceived benefits”

3. “Determining influence of service quality on user identification, belongingness, and satisfaction on mobile social media: Insight from emotional attachment perspective”

4. “How multidimensional benefits determine cumulative satisfaction and eWOM engagement on mobile social media: Reconciling motivation and expectation disconfirmation perspectives”

5.“Struggling or Shifting? Deciphering potential influences of cyberbullying perpetration and communication overload on mobile app switching intention through social cognitive approach”

6. “Determining the influence of depressive mood and self-disclosure on problematic mobile app use and declined educational attainment: Insight from stressor-strain-outcome perspective”

7�“Deciphering dynamic effects of mobile app addiction, privacy concern and cognitive overload on subjective well-being and academic expectancy: The pivotal function of perceived technostress”

Response: Thank you for sending details of these 7 studies, all of which have a single author in common.

These studies are not directly relevant to our paper. One is about an AI-powered Chatbot, the others relate to a range of issues as mobile social media, cyberbullying, and technostress.

We have discussed these papers as a research team and agreed that they are not directly relevant to the topic area of this paper and would detract from its focus.

Third, more details about the method are needed to add in this current article. Language needs polishing: many sentences are not very smooth.

Response: Thank you for this comment. We have made some minor revisions to sentences throughout the paper (particularly the methods) to improve the flow. We noticed inconsistency in the presentation of well-being (well-being or wellbeing) and this has been corrected throughout.

It is not clear what further methodological details are required by this reviewer – we have already included all relevant details for this study. One change we did make is to move the signposting to the interview topic guide (S1 Text) from the results section into the methods.

Fourth, according to the results, some findings were proposed. Combined with related work, the authors should accentuate their contributions in the conclusion section of this paper.

Response: Thank you for the opportunity to further accentuate our findings. We have endeavoured to do this with a re-draft of text in the conclusion section.

Finally, while the paper is well-structured, there are still some issues with the clarity of expression and language, undermining its readability. Thus, the authors should ensure that a revised version of the manuscript is thoroughly proofread.

Response: Thank you for this comment. We have re-read the manuscript and made minor revisions to the language used throughout.

Reviewer 2:

1. Methodological Considerations

Technical Issue Specificity: In Theme 3: Barriers to engagement – Technical Challenges, the mention of “software bugs” is too general. Since the app is central to the study, it would be helpful to describe the types of bugs encountered and their impact. Login issues on Android and the lack of a password reset feature appear to have contributed to attrition. Given that the app was updated during the study, please clarify whether these problems were resolved and why participants were not encouraged to update, as updates typically address such usability concerns.

Response:Many thanks for these helpful comments. In Theme 3, “unresponsive questions, delayed saving of data, and daily scores not loading” are the minor bugs. The next sentence (line 391) was revised, those bugs (occurred a few times) were gone after restarting the app. These bugs may also occur based on the phone quality.

There is one major technical issue with Android phones which is mentioned in line 231. This issue occurred during the recruitment period. We contacted the app developer, but unfortunately the issue could not be resolved. Since we had already reached our target number of participants, recruitment stopped.

Engagement Calculation: The reported 64.41% engagement rate (514/789 days) lacks detail. Please clarify how this was calculated over the 6-week period, including definitions of engagement and the denominator used.

Response: Mobile engagement means frequency of mobile app use, which is someone entering the app and answering the daily questions. We had 19 participants in the intervention group. It was expected all participants would use the app every day, total of 42 days. So, 19x42= 798 which is the total days. However, some participants used the app every day, but some only used it for a few days. When somebody used the app, the app saved the day and their daily score. Based on these records, we calculated the total actual use which was 514. Then, we divided 514/798 =64.4% that is the daily engagement rate. In the method section line 186-189, a sentence was added to describe how the engagement rate was calculated.

While re-checking the numbers, we noticed that there was a mistake relating to the total number. It is 798 but was mistakenly reported as 789. This has been corrected and all other numbers have been checked and are correct.

Tracking Frequency: Please specify how often participants used the app each day. This data is often available from app analytics. Encouraging more frequent use may have yielded richer insights, as prior research (e.g., Myin-Germeys et al., 2009) shows the importance of capturing intra-day fluctuations in mental health using Ecological Momentary Assessment (EMA).

Response: Many thanks for this suggestion. However, since participant were encouraged to use the app once per day for daily monitoring, we did not observe more than one use in a day and would not expect to in this particular study.

2. Inconsistencies

Usability Ratings: The usability section rates ease of use as “excellent,” yet interface satisfaction and usefulness are rated “poor.” This contradiction should be clarified.

Participant Feedback vs. Quantitative Ratings: In Theme 1, participants are quoted as praising the app’s design and ease of use, yet this conflicts with the MAUQ ratings. Clarifying this discrepancy would improve interpretive consistency.

Response: Thank you for this comment. This MAUQ included 18 questions, which were divided into three groups: ease of use (the first five questions), interface and satisfaction (the following seven questions), and usefulness (the last six questions). Participants expressed different views on these three groups.

The findings from the MAUQ are consistent with the qualitative findings. In the themes, the participants like the ease of use of the app. However, they indicate some cons about interface and satisfaction such as “masculine interface” (in Theme 4, design improvements) and perceived lack of usefulness (in Theme 1, Impacts on mental wellbeing). The MAUQ total score reflects these differences in views as its “good” in between “excellent” and “poor”.

Dropout and Well-being Scores: The "assessment of mental well-being, depression, stress and anxiety" section mentions that participants who dropped out had higher baseline well-being scores than those who completed the study. This pattern appears to undermine the conclusion that the app supports positive behavior change or improved well-being, and should be discussed.

Response: Many thanks for this thoughtful comment. However, high well-being score is better mental wellbeing based on the interpretation of the scoring of WEMWBS. In the intervention group, post-intervention mental well-being score is slightly lower than baseline mental-wellbeing score. It is also lower than the control group scores. So, this may mean that the app may not show significant improvement in the well-being scores. We have reflected on this in the discussion (line 446-459), although, as we have now added to the text, it is not the intention of a feasibility study to examine / determine effectiveness of an intervention. Small sample size, and higher well-being scores in drop-outs may impact on these differences.

3. Reporting Clarity

Repetition: The phrase “expectation of immediate benefits” is repeated in Theme 3 – Motivational Challenges and could be consolidated.

Comparative Claims: The claim that engagement was higher than in similar studies needs supporting references to validate the comparison.

Response: Yes, thank you for noticing this. The repetition has now been removed.

We appreciated the suggestion to cite references for our comparison with other studies relating to engagement. We have already cited two articles in which the sample is a healthcare population.

Knill K, Warren B, Melnyk B, Thrane SE. Burnout and well-being: Evaluating perceptions in bone marrow transplantation nurses using a mindfulness application. Clin J Oncol Nurs. 2021;25: 547–554. doi:10.1188/21.CJON.547-554

Orosa-Duarte Á, Mediavilla R, Muñoz-Sanjose A, Palao Á, Garde J, López-Herrero V, et al. Mindfulness-based mobile app reduces anxiety and increases self-compassion in healthcare students: A randomised controlled trial. Med Teach. 2021;43: 686–693. doi:10.1080/0142159X.2021.1887835

To further address your suggestion, we have now added a recent meta-analysis of persuasive design, engagement, and efficacy in 92 RCTs of mental health apps to support our argument that our engagement rate is higher than overall daily app use.

Valentine L, Hinton JDX, Bajaj K, Boyd L, O’Sullivan S, Sorenson RP, et al. A meta-analysis of persuasive design, engagement, and efficacy in 92 RCTs of mental health apps. NPJ Digit Med. 2025;8: 229. doi:10.1038/s41746-025-01567-5

The study explores an important area of interest. Addressing the issues above will improve the manuscript’s clarity, methodological transparency, and alignment between data and conclusions.

Response: Many thanks for your supportive comments which we believe have improved the manuscript quality.

Tracking Frequency: Please specify how often participants used the app each day. This data is often available from app analytics. Encouraging more frequent use may have yielded richer insights, as prior research (e.g., Myin-Germeys et al., 2009) shows the importance of capturing intra-day fluctuations in mental health using Ecological Momentary Assessment (EMA). Many thanks for these helpful comments. In Theme 3, “unresponsive questions, delayed saving of data, and daily scores not loading” are the minor bugs. The next sentence (line 391) was revised, those bugs (occurred a few times) were gone after restarting the app. These bugs may also occur based on the phone quality.

There is one major technical issue with Android phones which is mentioned in line 231. This issue occurred during the recruitment period. We contacted the app developer, but unfortunately the issue could not be resolved. Since we had already reached our target number of participants, recruitment stopped.

Mobile engagement means frequency of mobile app use, which is someone entering the app and answering the daily questions. We had 19 participants in the intervention group. It was expected all participants would use the app every day, total of 42 days. So, 19x42= 798 which is the total days. However, some participants used the app every day, but some only used it for a few days. When somebody used the app, the app saved the day and their daily score. Based on these records, we calculated the total actual use which was 514. Then, we divided 514/798 =64.4% that is the daily engagement rate. In the method section line 186-189, a sentence was added to describe how the engagement rate was calculated.

While re-checking the numbers, we noticed that there was a mistake relating to the total number. It is 798 but was mistakenly reported as 789. This has been corrected and all other numbers have been checked and are correct.

Many thanks for this suggestion. However, since participant were encourage to use the app once per day for daily monitoring, we did not observe more than one use in a day and would not expect to in this particular study.

2. Inconsistencies

Usability Ratings: The usability section rates ease of use as “excellent,” yet interface satisfaction and usefulness are rated “poor.” This contradiction should be clarified.

Participant Feedback vs. Quantitative Ratings: In Theme 1, participants are quoted as praising the app’s design and ease of use, yet this conflicts with the MAUQ ratings. Clarifying this discrepancy would improve interpretive consistency.

Dropout and Well-being Scores: The "assessment of mental well-being, depression, stress and anxiety" section mentions that participants who dropped out had higher baseline well-being scores than those who completed the study. This pattern appears to undermine the conclusion that the app supports positive behavior change or improved well-being, and should be discussed. Thank you for this comment. This MAUQ included 18 questions, which were divided into three groups: ease of use (the first five questions), interface and satisfaction (the following seven questions), and usefulness (the last six questions). Participants expressed different views on these three groups.

The findings from the MAUQ are consistent with the qualitative findings. In the themes, the participants like the ease of use of the app. However, they indicate some cons about interface and satisfaction such as “masculine interface” (in Theme 4, design improvements) and perceived lack of usefulness (in Theme 1, Impacts on mental wellbeing). The MAUQ total score reflects these differences in views as its “good” in between “excellent” and “poor”.

Many thanks for this thoughtful comment. However, high well-being score is better mental wellbeing based on the interpretation of the scoring of WEMWBS. In the intervention group, post-intervention mental well-being score is slightly lower than baseline mental-wellbeing score. It is also lower than the control group scores. So, this may mean that the app may not show significant improvement in the well-being scores. We have reflected on this in the discussion (line 446-459), although, as we have now added to the text, it is not the intention of a feasibility study to examine / determine effectiveness of an intervention. Small sample size, and higher well-being scores in drop-outs may impact on these differences.

3. Reporting Clarity

Repetition: The phrase “expectation of immediate benefits” is repeated in Theme 3 – Motivational Challenges and could be consolidated.

Comparative Claims: The claim that engagement was higher than in similar studies needs supporting references to validate the comparison.

The study explores an important area of interest. Addressing the issues above will improve th

---

## [Decision Letter · Decision Letter 1]

17 Nov 2025

Dear Dr. Blake,

Thank you for submitting your manuscript to PLOS ONE. After careful consideration, we feel that it has merit but does not fully meet PLOS ONE’s publication criteria as it currently stands. Therefore, we invite you to submit a revised version of the manuscript that addresses the points raised during the review process.

I’m glad to hear from the reviewer that the paper has improved!

Could you please specify the minor issues you’d like me to address? If you can list them by the relevant parts, I’ll fix everything right away.

We look forward to receiving your revised manuscript.

Kind regards,

Najmul Hasan, PhD

Academic Editor

PLOS ONE

Journal Requirements:

Reviewers' comments:

Reviewer's Responses to Questions

**Comments to the Author**

Reviewer #2: All comments have been addressed

Reviewer #3: All comments have been addressed

2. Is the manuscript technically sound, and do the data support the conclusions?

Reviewer #2: Yes

Reviewer #3: Yes

3. Has the statistical analysis been performed appropriately and rigorously?

Reviewer #2: Yes

Reviewer #3: Yes

4. Have the authors made all data underlying the findings in their manuscript fully available?

Reviewer #2: Yes

Reviewer #3: Yes

5. Is the manuscript presented in an intelligible fashion and written in standard English?

Reviewer #2: Yes

Reviewer #3: Yes

Reviewer #2: Thank you for addressing the comments and providing details.

The manuscript shows improvement with stronger evidence and explanations. However, a few points still need clarification:

Line 231 – Android Bug: Please specify the exact “major Android bug” mentioned. It’s unclear what issue this refers to and how it affected the study. It is important to understand the technical difficulties as the app is the primary tool.

App Update: Clarify whether the app update during the study resolved the usability problems and why participants were not encouraged to update, since updates often fix such issues (especially the major Android bug mentioned above).

Overall, the paper is well improved and close to readiness once these clarifications are addressed.

Reviewer #3: The manuscript is conceptually strong and methodologically appropriate for a feasibility study, but requires clarifications in methods, moderating of claims, and more precise framing of results.

**Do you want your identity to be public for this peer review?** For information about this choice, including consent withdrawal, please see our Privacy Policy

Reviewer #2: No

Reviewer #3: **Yes:**  Asma Khaleel Abdallah

---

## [Author Response · Author response to Decision Letter 2]

17 Nov 2025

17 November 2025

Dear Editor,

Thank you for the opportunity to revise and resubmit our article:

[PONE-D-25-19125R1] - [EMID:cbd3df42c2f44951]. A point-by-point response is provided below.

We look forward to hearing from you in due course.

Kind regards,

Prof. Holly Blake (corresponding author).

Comments Revisions

Reviewer 2

All comments have been addressed Thank you for confirming that all comments have been addressed to satisfaction.

Reviewer has confirmed YES to:

The manuscript is technically sound, and the data support the conclusions.

Statistical analysis been performed appropriately and rigorously.

All data underlying the findings in the manuscript are fully available.

The manuscript presented in an intelligible fashion and written in standard English.

The manuscript shows improvement with stronger evidence and explanations. However, a few points still need clarification:

Line 231 – Android Bug: Please specify the exact “major Android bug” mentioned. It’s unclear what issue this refers to and how it affected the study. It is important to understand the technical difficulties as the app is the primary tool.

App Update: Clarify whether the app update during the study resolved the usability problems and why participants were not encouraged to update, since updates often fix such issues (especially the major Android bug mentioned above).

Overall, the paper is well improved and close to readiness once these clarifications are addressed. Thank you for this positive feedback.

In response to the two comments:

The major Android bug, referenced on line 231, was a persistent authentication error. Even though new Android users entered correct credentials, they received "invalid password or email" error resulting no access to the app. This issue was not seen in IOS users.

Regarding the app update, the developer informed us that due to financial constraints resulting from the COVID-19 pandemic, they were unable to develop or release a patch to resolve this specific major bug and other minor bugs. Therefore, no update was available to offer participants. However, minor bugs faced by a few participants were gone after restarting the app.

We have faced this issue after recruiting 49 participants which was a sufficient number for a feasibility study (e.g. Totton et. al., 2023). Therefore, since we reached the sufficient participant number for our study, and were unable to resolve this major issue, we did not recruit any further participants to the study.

The relevant section has been revised in the manuscript.

Reviewer 3

All comments have been addressed Thank you for confirming that all comments have been addressed to satisfaction.

Reviewer has confirmed YES to:

The manuscript is technically sound, and the data support the conclusions.

Statistical analysis been performed appropriately and rigorously.

All data underlying the findings in the manuscript are fully available.

The manuscript presented in an intelligible fashion and written in standard English.

The manuscript is conceptually strong and methodologically appropriate for a feasibility study, but requires clarifications in methods, moderating of claims, and more precise framing of results. The review has confirmed that:

1. All comments have been addressed (in the revised version previously submitted).

2. The manuscript is conceptually strong.

3. The methodology is appropriate for a feasibility study.

The reviewer has now made an additional comment that this manuscript “requires clarifications in methods, moderating of claims, and more precise framing of results” but has not given any indication as to what should change.

We have already been very precise in our reporting of methods and results, which is guided by the Consolidated Standards of Reporting Trials (CONSORT) for pilot and feasibility studies, the Checklist for Reporting Qualitative Studies (COREQ-32), and mobile health (mHealth) evidence reporting and assessment (mERA) checklist. Therefore, we do not believe there is anything further to clarify in the methods, except for addressing the two specific comments about the App, made by Reviewer 2.

We have not overstated our claims, and believe the interpretation is appropriate in line with reporting guidelines for feasibility studies (Consolidated Standards of Reporting Trials (CONSORT) for pilot and feasibility studies). We conclude that “Our study is the first to examine the feasibility and detailed daily usage of a well-being monitoring mobile app for mental well-being promotion among healthcare workers and trainees. This study demonstrates the feasibility of a mental well-being monitoring mobile app for healthcare workers and trainees, with positive findings related to the acceptability of randomised controlled trial design methodology including recruitment, randomisation, intervention delivery, questionnaire implementation, and participant retention”. Using cautious language, we indicate that “Although a feasibility study does not aim to examine effectiveness of the intervention…” and “the effectiveness of such interventions needs to be a tested in a large-scale randomised trial”.

However, we have re-drafted the whole study strengths and limitations section, and sections of the results, discussion and conclusion to ensure maximum clarity.

Note that this reviewer has confirmed that “all comments have been addressed”, “The manuscript is technically sound, and the data support the conclusions”, “Statistical analysis been performed appropriately and rigorously”, “All data underlying the findings in the manuscript are fully available” and “The manuscript presented in an intelligible fashion and written in standard English”.

---

## [Decision Letter · Decision Letter 2]

16 Dec 2025

Dear Dr. Blake,

Thank you for submitting your manuscript to PLOS ONE. After careful consideration, we feel that it has merit but does not fully meet PLOS ONE’s publication criteria as it currently stands. Therefore, we invite you to submit a revised version of the manuscript that addresses the points raised during the review process.

We look forward to receiving your revised manuscript.

Kind regards,

Najmul Hasan, PhD

Academic Editor

PLOS One

**Journal Requirements:**

Reviewers' comments:

Reviewer's Responses to Questions

**Comments to the Author**

Reviewer #2: All comments have been addressed

2. Is the manuscript technically sound, and do the data support the conclusions?

Reviewer #2: Yes

3. Has the statistical analysis been performed appropriately and rigorously?

Reviewer #2: Yes

4. Have the authors made all data underlying the findings in their manuscript fully available?

Reviewer #2: Yes

5. Is the manuscript presented in an intelligible fashion and written in standard English?

Reviewer #2: Yes

**Reviewer #2:** All comments have been addressed in the latest submission. The Android problem specifically is also added.

**Do you want your identity to be public for this peer review?** For information about this choice, including consent withdrawal, please see our Privacy Policy

Reviewer #2: No

---

## [Author Response · Author response to Decision Letter 3]

22 Dec 2025

17 December 2025

Dear Editor and Reviewers,

PONE-D-25-19125R2

Supporting mental well-being of healthcare workers using a mobile app: a mixed-methods feasibility study

We have received a request for minor revisions to the above paper. However, there do not seem to be any revisions to address.

We have responded to the points made below which confirm this. Consequently, no changes have been made to the documents submitted last time.

If you would like to make changes to your financial disclosure, please include your updated statement in your cover letter

We have no changes to make, the information already included is correct, although we have included it again in our cover letter.

Comments to the Author

1. If the authors have adequately addressed your comments raised in a previous round of review and you feel that this manuscript is now acceptable for publication, you may indicate that here to bypass the “Comments to the Author” section, enter your conflict of interest statement in the “Confidential to Editor” section, and submit your "Accept" recommendation.

Reviewer #2: All comments have been addressed

Thank you for this positive comment.

2. Is the manuscript technically sound, and do the data support the conclusions?

Reviewer #2: Yes

Thank you for confirming our manuscript is technically sound.

3. Has the statistical analysis been performed appropriately and rigorously?

Reviewer #2: Yes

Thank you confirming the statistical analysis has been performed appropriately and rigorously.

4. Have the authors made all data underlying the findings in their manuscript fully available?

Reviewer #2: Yes

Thank you for confirming our data are fully available in a research data management repository.

5. Is the manuscript presented in an intelligible fashion and written in standard English?

Reviewer #2: Yes

Thank you for confirming that our manuscript is written in an intelligible fashion and standard English.

6. Review Comments to the Author

Reviewer #2: All comments have been addressed in the latest submission. The Android problem specifically is also added.

Thank you for confirming that all comments were addressed in the previous submission.

We have checked and our figures already do meet technical requirements.

---

## [Editor Report · Decision Letter 3]

2 Jan 2026

Supporting mental well-being of healthcare workers using a mobile app: a mixed-methods feasibility study

PONE-D-25-19125R3

Dear Dr. Blake,

We’re pleased to inform you that your manuscript has been judged scientifically suitable for publication and will be formally accepted for publication once it meets all outstanding technical requirements.

Kind regards,

Najmul Hasan, PhD

Academic Editor

PLOS One
---

## [Editor Report · Acceptance letter]

PONE-D-25-19125R3

PLOS One

Dear Dr. Blake,

I'm pleased to inform you that your manuscript has been deemed suitable for publication in PLOS One. Congratulations! Your manuscript is now being handed over to our production team.

Kind regards,

on behalf of

Dr. Najmul Hasan

Academic Editor

PLOS One